# Social Innovation in Long-Term Care: Lessons from the Italian Case

**DOI:** 10.3390/ijerph17072367

**Published:** 2020-03-31

**Authors:** Georgia Casanova, Andrea Principi, Giovanni Lamura

**Affiliations:** IRCCS—INRCA—National Institute of Health & Science on Ageing, Centre for Socio-Economic Research on Ageing, 60124 Ancona, Italy; a.principi@inrca.it (A.P.); g.lamura@inrca.it (G.L.)

**Keywords:** social innovation, long-term care, Italy, ageing

## Abstract

The debate on policies addressing the challenges posed by population ageing pays increasing attention to sustainable and innovative ways to tackle the multidimensional impact this phenomenon has on society and individuals. Moving from the findings of two European research projects, a qualitative study based on a rapid review of the literature, expert interviews, focus groups and case studies analysis has been carried out in Italy. This study illustrates which social innovations have been recently implemented in this country’s long-term care (LTC) sector, and the areas in which further steps are urgently needed in the future. This takes place by first highlighting the existing links between social innovation and LTC, and then by identifying the key factors that can facilitate or hinder the implementation of these initiatives. Finally, the study suggests how to promote social innovation, by strengthening the “integration” and “coordination” of available services and resources, through a—for this country still relatively—new approach towards ageing, based on pillars such as prevention and education campaigns on how to promote well-being in older age.

## 1. Introduction

The Italian long-term care (LTC) system, as in most Mediterranean countries, is usually labeled as ‘familialist’, because the role of formal care provision is quite marginal compared to that played by the pervasive assistance traditionally granted by family members [1,2,3,4,5,6,7]. Some studies, however, have underlined a trend towards the “hybridization” of care in many European countries, including Italy, describing them as “mixed care regimes” [8,9,10]. Notwithstanding that, the risk of a high level of unmet needs remains one of the main challenges in the Italian LTC sector, also due to consistently low LTC expenditure, which has never exceeded 1.9% of the GDP over the last decade [11,12,13]. This is all the more worrying in the light of the continued ageing of the Italian population, and, therefore, of the overall need for care: in 2017, 22.6% of the Italian population was 65 or older, and the national ageing index reached a value of 168.9 [14]. Moreover, care services in the Italian LTC system have been recently decentralised from the national to the regional/local level, and are characterised by a high level of fragmentation, due to lack of integration and coordination strategies between the health and the social care sectors [15]. Not surprisingly, in the last decade in Italy, similarly to other European countries, the debate on LTC-related challenges has increasingly focused on the search for innovative solutions that pay attention to the quality of life of older people and of their carers [3,16,17,18,19]. Within this debate, social innovation (SI) has become one of the most recommended strategies to face the new social risks deriving from the structural changes in socio-demographics and societal needs, among which age-related care plays a predominant role [17,20]. The definition of SI provided by the European Commission suggests that this term refers to “any new idea—including products, services and models—that simultaneously meets social needs—more effectively than alternatives—and creates new social relationships or collaborations, i.e., it is both good for society and enhances society’s capacity to act” [21]. International debate in this field has identified three main LTC areas in which the impact of SI is at its most significant [22,23]: the search for appropriate governance approaches to promote effective home care; the methods and strategies to integrate formal provision—identified as remunerated care supplied by professionals or services—and informal care, defined as unpaid assistance granted by family members, friends or neighbours; and, lastly, the increasing presence of “migrant care workers”, legally or illegally employed by households with frail older members. Informal carers—along with professional carers —have, as a result, become a crucial and inevitable component of the most co-modified care regimes [24]. The above identified SI characteristics meet the general features of familistic care regimes, since recent studies show that SI in these countries is often implemented to improve the relationship between formal care and informal care provided at home [25,26,27].

In this respect Italy, as an example of a “familistic” care regime, has been drawing increasing attention from international observers as a potential incubator for innovative policies, in the light of a number of experiences highlighted by some recent European studies [27,28]. Since the bulk of care in Italy is provided primarily by families, it is not surprising that innovation in the LTC sector is more focused on home-based care [28]. In this regard, it should be underlined that the real “pillar” of Italy’s LTC is the help provided by migrant care workers, who represent about 80% of all privately hired household assistants [29]. Another fundamental element to understand Italy’s LTC is the decentralisation of health and social governance, including LTC, namely the regions, which are the main local institutions, identify strategies, policies and instruments, within the tangled web of the national regulation framework. In this regard, the widespread involvement of multiple stakeholders in governance models, which is frequently used in Mediterranean countries, promotes an intensive dialogue between different actors in the search for new integration and coordination solutions [15,30]. The Italian case gained an international relevance thanks to its recent, peculiar trends in LTC-related issues. The Italian LTC system is often analyzed in combination with other Mediterranean countries (e.g., Spain, Portugal, Israel), because here families remain the core providers of care for frail older people, as reflected by the label of “familistic” care regime [31,32,33,34]. On the other hand, however, a number of studies have stressed that the Italian LTC system also reports some similarities with countries with a more “mixed regime”, for its push—albeit weak—towards the integration and coordination of care services [35]. For these reasons, the Italian case represents an interesting case for the current debate on the spread of SI in other countries.

In the light of this context, this paper attempts to analyze the role played by SI solutions applied to the Italian LTC system. To this end, we are investigating the overall framework in which SI-LTC relationships are developed, identifying which SIs are more urgently needed to reform the system, as well as the key drivers that might best support the dissemination of innovative solutions. The multi-level analysis of the Italian LTC system aims also to identify the link between the investigated innovations and the different governance layers of policy implementation (macro, meso and micro), in order to understand which level policy suggestions and recommendations can best refer to, in an attempt to provide useful insights for non-Italian contexts as well. 

## 2. Materials and Methods

A qualitative analysis was carried out on data collected within the two projects Mobilising the Potential of Active Ageing in Europe [36] and Cost Effectiveness and Quality in long-term care [37]. This was achieved by means of a combination of focus groups, expert interviews and an analysis of good practices. A preliminary rapid review was also carried out to support the qualitative analysis. This approach was considered to be more efficient than a study based on a single qualitative tool to achieve the study aims [38]. 

As for the rapid literature review (for a recent overview of this methodology see Tricco et al. [36]) a limited set of keywords was identified for review purposes: “Social Innovation”; “Long-Term Care”, “Active Ageing” and “Italy”. While all other keywords are self-explanatory, the term “active ageing” was included because the literature underlines the strong linkage this concept has with LTC, in particular with regard to the disability-preventative power of healthy active ageing along the ageing process [35,39,40]. These keywords were used in combination, in order to reduce the number of documents with little or no relevance, and inserted as search items in the Google, Google Scholar, and Scopus databases. We found 204 English and Italian records, without duplicates, published from 2012 to 2019 that were screened by the eligibility criteria. The first author screened the identified studies for their relevance based on title and abstract. In particular, records were included if: (a) they had been published between 2012 and 2019; (b) they were focused on the Social Innovation definition and its main characteristics in Europe; (c) they were focused on innovative long term care solutions or active ageing policies and practices in Europe and, in particular, in Italy. Papers not focused on policy or practices and methodological papers were excluded. The selected articles were retrieved full text and reassessed by authors for final inclusion based on consensus. 22 papers were included in this rapid review (Figure 1). No other references were identified by hand searching or in the references of the included articles.

The rapid review allowed the authors to analyze how the SI and LTC concepts were used in both specialized and grey literature and identify the policy conditions that enable mere “practice” to be transformed into “social innovation practice”. In addition, an analysis of relevant policy documents and reports was performed, to pinpoint the practices to be included in the study. For this purpose, selection criteria were drawn from the SI definition and the findings of the previous literature review, referring in particular to Heinze and Naegele [41] and Kesselring et al. [42]. These criteria were based on key aspects of SI and LTC, such as the level of integration, the achieved impact, the addressed target(s), the potential for transferability and the long-term sustainability of initiatives.

Two focus groups and twelve semi-structured interviews with experts representing different stakeholders were conducted (Table 1). The data collection was carried out by the first author in the period between September 2015 and March 2018. This choice allowed the consolidation of experts’ opinions on the application of the SI concept—as defined by the European Commission—in LTC real life contexts. The 32 experts shown represented circa 95% of all the experts initially contacted (as circa 5% of those asked could not or did not agree to take part in the study within its time frame). The time needed to realize the interviews reached on average circa 60 min, while focus groups lasted no longer than 160 min in total.

The experts were selected according to the relevance of their academic or professional profiles, as reflected by their participation in the national—and sometimes even international—debate on LTC related issues. In order to consider different perspectives, participants were selected by using a mixed strategy, representing the following organizations: local institutions, especially Regions and Municipalities; non-profit or profit organizations; private sector care providers; trade unions; foundations specializing in research on health and social issues; Information and Communication Technologies (ICTs) industries in the health care sector; research centers on ageing, welfare and labor market; individual experts in these fields. Considering the large differences existing within Italian regions in LTC provision and welfare mix, the focus groups covered two different local contexts: one in Milan, the capital city of the Lombardy region and largest city in the northern part of the country, often labeled as the “Italian business capital”; and one in Ancona, a medium-sized city in central Italy, representing the capital of the Marche region. In both cases, a local observational perspective was used, thus leveraging the analysis of the local situation as a starting point to develop the discussion for the specific focus group. The focus groups and expert interviews focused on the following issues:-What is social innovation?-How much room for social innovation exists in the Italian LTC system?-What innovations in LTC do you consider to be most urgent in this country?-What is the current situation in this respect?-Are there examples of these specific social innovations already implemented in Italy?-What are the key factors, defined as barriers and drivers, for the implementation of the identified social innovations?-What are the main recommendations for Italian and international policymakers?

As for the data analysis, we used the conceptual framework based on Braun and Clarke’s Six Phases of Thematic Analysis [43]. For coding purposes, a deductive approach was followed that aimed at reducing the complexity of data and linking the research questions to the collected data [44].

The three Italian good practices included in the study were labeled as “social innovation case studies” in the MoPAct project and subsequently further analyzed in the Cequa Project. The method used to select the cases was defined by means of a common agreement between the eight international teams working in the MoPAct project, including the first author of this study. Each European experience collected by national teams was awarded a score by the other teams involved in the study. A score from 1 to 5 was assigned to measure the presence (low/high) of each one of six dimensions required to be labelled “SI good practice” [29]: key aspects for integrating LTC status (concluded/ongoing); impact (direct user/society); transferability; sustainability; and key “conditions” for SI (as categorized by Heinze and Naegele [41] and detailed in Table 2). Experiences were finally assessed according to the sum of the above-mentioned scores: the higher the score, the more innovative was the experience considered, and identified as “good practice”. The analysis of case studies has been presented to the focus groups and during interviews as examples of SI, to support the discussion. The experts have been called to give their agreement on whether the selected case studies could be considered good practices, and whether the chosen criteria to identify them were appropriate for the Italian context. Finally, the spokespersons of the selected experiences also participated in the focus groups or were interviewed as stakeholders.

## 3. Results

Findings were grouped in accordance with the study goals and illustrated separately in four subsections: the relationships between SI and LTC; the main social innovation demand in Italy; the key factors currently supporting/preventing Social Innovation; and policy recommendations. 

### 3.1. The Relationship between Social Innovation and LTC

The analysis of the literature underlined, first of all, that in Italy the concept of SI is seldom used to label initiatives and policies. In many of the contributions, this topic is introduced only indirectly, stressing its role as a factor that can promote and drive changes in the Italian care system. The topic of SI in LTC was invoked more directly by some “grey literature” papers and in some experimental experiences, often related to European projects. Secondly, the literature review showed that SI was often intended as a “technological innovation” in LTC (such as, for instance, ambient assisted living tools, domotic solutions, ICT-based services, etc.). Although it is certainly true that new technologies can promote innovation, and sometimes SI, too, the overwhelming attention given to technologies as drivers of SI seemed to reflect the current difficulty in understanding this concept in its different, multi-faceted aspects. Not surprisingly, most of the literature in this regard that was found for the Italian context directly referred to the use of ICT in LTC [45].

The results that emerged from the focus groups and interviews confirmed that the concept of SI was not yet well established among the majority of Italian stakeholders. These often had only an approximate idea initially of what the general definition proposed by the European Commission [3] meant and gained a clearer understanding only when it was directly correlated to practical LTC activities or other ageing-related issues. As for the relationship between the concepts of SI and LTC, both concepts had three common objectives: (a) the improvement of individuals’ psycho-physical health and well-being; (b) the optimization of opportunities at both micro (individual) and macro (societal) level; (c) improvement of the quality of life in older age (meant in a holistic way, beyond mere health-related aspects, and including also social, economic and other dimensions that impact on quality of life). 

Based on these first results, the following related question was posed to both focus groups and experts: “*How much room for social innovation exists in the Italian LTC system*?” The consulted stakeholders underlined the emerging vision of SI as a strategic element for the future of Italian LTC, especially for the improvement of the care delivered in non-residential settings. The experts stressed furthermore that the promotion of SI is “necessary and urgent” in home care, a common feeling particularly strong among stakeholders from Italy’s southern regions. 

Focus groups and interviews helped to bring about an understanding of which LTC issues would be affected by the contribution from SI. Four main different areas were identified:Perception of “old age: in this respect, both focus groups and experts highlighted that it would be helpful to stop thinking of “older age” as a fixed segment of life starting at 65, and rather start referring to “ageing” as a more or less gradual process throughout the whole life. This would imply a change in policy vision, shifting from an approach aimed only at covering needs to policies designed instead at preventing them. According to this vision, the population-target is not only restricted to older people, but includes healthy adults and the whole life course. The experts agreed on how prevention policies would help us drive citizens to change their approach towards the risk of becoming dependent, thus prompting them to be more active early on in life by adopting healthy lifestyles and behavior;Efficiency of the care system: in this regard, the promotion of SI was identified as a fundamental strategy to achieve a more effective use of available resources, in terms of both formal and informal care provision;Partnership among stakeholders: with regard to this issue, it was emphasized that the adoption of a more “inclusive” approach towards stakeholders’ participation would be beneficial. In this respect, the definition of stakeholders provided by experts was particularly relevant: informal carers, migrant care workers and care recipients—often still neglected and marginalized—were considered as a fundamental part of the LTC system, since they were “partners” in everyday care delivery;Integration of services: in this area, SI was considered as a potential driver. In particular, innovative technologies have been easily identified for their contribution to new integrated home care paths. In addition, but less frequently, the experts labelled the organizational improvements directed at promoting integration and coordination of services (i.e., coordinate governance; case management, etc.) as SI.

In the end, the focus groups and interviews stressed the widespread presence of already existing SI practices around Italy and Europe. For example, the multi-stakeholder networks supporting the governance of specific local LTC initiatives are deeply rooted, but not well known outside of the circle of insiders directly involved with them. The experts underlined once more that the existing practices can be useful in building broader experiences.

The analysis of the three selected Italian good practices confirmed the strong links observed between high quality LTC and SI. Table 2 identifies the SI conditions proposed by Heinze and Naegele [41] that were considered for each of the three case studies, which are characterized by four, five and seven SI conditions, respectively. This extent is well above the minimum level of two considered as a threshold for an experience to be valued as socially innovative [26]. Two conditions were common to all three good practices: the presence of “*suggestions for new solutions in the respective societal, cultural and economic context*”; and the “*promotion of the integration and collaboration/partnership of heterogeneous stakeholders that have hitherto not co-operated*”. By contrast, the only condition that was not found at all concerned the “*involvement of end-users as co-producers of services and products*”, an area which seems to be neglected and would therefore require more attention in future efforts to improve SI in this sector in Italy.

**Table 2 ijerph-17-02367-t002:** Italian good practices and conditions of social innovation.

Practices Selected	Description		Social Innovation Conditions *								
			a	b	c	d	e	f	g	h	i
Family Nurse Programme	The private company Finisterre consortium provides health care services in many local health districts (LHD’s) in the Lombardia Region. The Family Nurse Programme idea is to create a group of LTC professionals able to more effectively respond to the various care needs related to chronic health conditions. In 2013 the Family Nurse Programme was placed on the Saronno district. The main activities of the family nurse are: direct hands-on assistance to the patient in his/her home through technical nursing interventions and rehabilitation; health education and provision of information to raise the awareness of the patient and his/her primary care network about specific requirements and options pertaining to the patient’s case; technical support to help family members in managing the patient independently. The family nurse collaborates with municipal social services and with primary care physicians, thereby creating a network of interdisciplinary care.		x	x	x		x				
Recognition of informal skills	Having worked for a long time in the care sector without any structured training, informal carers and care workers have often nevertheless accumulated important skills that are also required in professional care. Since 2008 the Piedmont Region promoted a programme focused on recognition and certification of care workers skills and training programmes, and activities to support increasing formal employment contracts. The initiative has been created to enhance the effectiveness of home based care that is carried out by privately paid migrant care workers. The programme includes a tools assess to identify (informally acquired) skills and to establish services to support and organise this process through certification and mentoring. In addition to the skills recognition component, the programme also supports local networks of services for people with home care needs. This includes information, reception and orientation training, matching demand for home care with skilled carers and helping with the administrative management of employment contracts. Lastly, the programme also includes subsidies to households for temporary replacements of carers in training.		x	x	x		x	x	x		x
Up-Tech	This project aims to reduce the burden on family carers of older Alzheimer’s Disease (AD) patients in order to allow the patients to live at home for as long as possible. It is currently being implemented in five health districts of the Marche Region in Italy. The targets are AD sufferers and their carers. The main components are: (a) case manager (an ad hoc trained social worker); (b) use of second generation telecare devices at home; (c) establishment of a collaborative working group [46]. Participants are divided into three groups: the first receives comprehensive care and support from the case manager; the second is similarly supported by a case manager, but receives also a technological toolkit. In order to control the effectiveness of these solutions, a third group of participants receives only information brochures regarding available services. All subjects are visited at home by a trained nurse who assesses them using a standardized questionnaire at enrollment (M0), 6 months (M6) and 12 months (M12). Follow-up telephone interviews are performed at 24 months (M24). The primary outcomes are: (1) caregiver burden, measured using the Caregiver Burden Inventory (CBI); and (2) the actual number of days spent at home during the study period, defined as the number of days free from institutionalizations, hospitalizations and stays in an observation unit of an emergency room [47].			x		x	x	x			x
* Legend based on [41]: a. It is oriented towards exceptional societal challenges/social issues; b. It suggests new solutions in the respective societal, cultural and economic context; c. It creates new patterns of social practices to overcome shortcomings of traditional arrangements; d. It tends to overcome the traditional dichotomy between technological and social innovations;		e. It promotes the integration and/or collaboration/partnership of heterogeneous stakeholders that have hitherto not co-operated; f. It includes reflective and multidisciplinary approaches towards the key goal of societal usefulness; g. It creates structures and processes that are sustainable and realize new growth potentials in terms of regular employment; h. It involves end-users as co-producers of services or products; i. It creates new roles and partnerships.									

### 3.2. Most Urgently Needed Social Innovations 

According to what reported above, the limited reference to the concept of SI in the literature can be attributed to the peculiar debate that is taking place on future changes in the Italian LTC system. The SI initiatives found in the grey literature were discussed as examples of good practice, but discussed less as urgent innovations to be spread to promote a structural change. In this regard, more details could be gained thanks to the experts’ opinions. In response to the question on what kind of innovations might be the “most urgent” in the current Italian LTC context, the experts identified seven main innovations, which are summarized in Table 3. An in-depth analysis of these innovations has revealed that they can be grouped into three main categories:Innovations addressing user needs;Innovative organizations and governance models;Innovations in the field of prevention.

Under the first category, social innovations are characterized by a particular focus on user needs. User’ needs are the starting point both to deliver personalized care—as recalled by the “taking on of case” concept—and for the integration of professionals and services, as proposed by the case management strategy. The SI labeled as “integrated mixed services for giving support to family care” focuses its attention on both the care worker’s needs (e.g., training, tutoring, etc.) and the preferences expressed by the care recipients and their families in their role of service users (e.g., choice of care worker, support for administrative aspects, etc.). This innovation contributes to an expansion of the concept of “user”, by including under this term the families of both care recipients and care workers (currently neglected by most service providers). 

The second category pointed to the fact that SI can be promoted by governance choices provided by new organizations or forms of collaboration. This category of innovation aims to bring out the innovative power in LTC from the involvement of different point of views and stakeholders. 

Focusing attention on multi-stakeholder networks, differences can be observed across Italy, mainly concerning ways of managing specific health and social support care services at a local level. The interviews underlined that these networks meet the declared SI aim “to promote new social relationships or collaborations”. Moreover, innovation is increased by valuing every single stakeholder and its contribution towards achieving the aims of declared LTC initiatives and policies, by cooperating effectively and efficiently.

On the other hand, the experts identified under the term “agencies to promote innovation in LTC and welfare system” a promising new governance approach to promote SI, extending to private operators the responsibility for disseminating innovation. In particular, according to the experts’ opinions, insurance companies and banks might become crucial SI drivers, e.g., by offering specific, innovative insurance schemes to cover the dependency risk. In this regard, an important role is played by private foundations—including banking foundations—thanks to the funding of specific initiatives or policies. Indeed, the “calls for proposals” by these foundations have become widespread practice across the country. Beyond single initiatives, however, the establishment of an “agency for innovation” might be a more promising strategy to promote SI on a large scale, as suggested by a national expert with a specific expertise on southern Italian regions. This idea is based on the potentially attractive and multiplier nature of such agencies to stimulate the production of innovation. The expert argued that these innovation agencies create SI within the care system by stimulating the conception of new ideas and/or new strategies, and by facilitating local stakeholders’ relationships. Moreover, one of their main objectives should be the promotion of skills building and of new ideas around LTC and the issue of an ageing population, in the attempt to lead to better solutions to clearly identified local needs. 

Finally, according to experts in Italy SI is urgently required in the field of prevention (third category). At first, we encountered contradictory opinions as to who, public or individual, should bear the greatest responsibility for prevention. The assumption of a public vision supported the idea that extending the provision of care services is a way of implementing innovative solutions; on the other hand, individual responsibility involved the role of citizens and their individual choices, in which case SI should be focused on a “cultural” change of citizens’ habits and choices. 

In the focus groups, the experts tried to find a balance between these visions, identifying two main SIs oriented to prevention, labeled as “pro-active prevention” and “education on dependency”.

Pro-active prevention, a SI suggested mainly by the Milan focus group, is based on the idea that older people should be prepared for a possible future condition of dependency. The experts emphasized that public authorities should maintain a leading role in this regard, and that this should be carried out by a collaboration between services, institutions and formal stakeholders. However, the debate brought to light the fact that, in Italy, prevention remains mostly an individual choice. The examples of prevention activity mentioned were mainly information campaigns, rather than real in-kind services or policies.

A partially different approach was proposed by the Ancona focus group, which prompted a vision of “prevention” based on educating the population on how to best prepare for the ageing process, including the possibility of having to face potentially long periods of dependency. The main aim of education on dependency is in this case the improvement of people’s awareness, by addressing families, schools, mass media and other institutional stakeholders.

Both prevention innovations emerged from a debate showing that in Italy ageing is still seen in the eyes of large sections of the population as an “uncomfortable thing” or as “something far from us”.

### 3.3. Key Factors to Promote Social Innovation Actions

From the literature reviews, three SI key factors emerged as most required in the Italian LTC. The first refers to the implementation of strategies that address older people’s quality of life. By addressing this goal explicitly, older people become the key focus in creating SI and care services should be improved in order to increase their responsiveness to recipients’ needs [48,49]. Secondly, the literature proposed extending the definition of users to care workers (migrant care workers included), as this would help promote the welfare of the whole professional community [50]. Lastly, the literature underlined the importance of stakeholder involvement in the care system, identifying two levels: the first consisting of formal actors (i.e., institutions, voluntary associations, private subjects, etc.), and the second including stakeholders associated with older people (i.e., family carers, privately employed care workers, etc.), whose role within the care system must still be enhanced.

The evidence that emerged from the focus groups and interviews confirmed the findings from the literature review. The experts agreed in particular on the overall scheme of key factors promoting SI, as proposed by the model offered by Goldman and colleagues [51], also adopted by the MoPAct research team [26]. As for the focus groups, the consulted experts had been invited to identify the meaning of each key factor for the Italian LTC context. Table 4 summarizes the results of this consultation, which highlighted the following as the three most important ones:*framework/structural conditions*: these factors referred mainly to the optimization of existing resources (e.g., redistribution of public expenditure and reform of the LTC system);*local/community focus*: when planning policies, focus was given to local needs;*network and leadership*: this includes, on the one hand, an increased responsibility taken on by the third sector (represented by voluntary organizations and NGOs) and, on the other hand, a more widespread use of co-planning, involving different typologies of stakeholders and actors

### 3.4. Policy Recommendations

The recommendations received from the experts and stakeholders consulted are related to the results discussed above. The most widely shared recommendation was that of soliciting a national LTC reform, because this would allow SI to found different forms of implementation. To achieve this, the national LTC fund has to become permanent and to be increased in terms of resources, thus transcending the current annual planning via the national tax law. Such a reform should furthermore consider a series of additional recommendations, summarized as follows:(1)To promote national policies directed at organizing the care services system, based on SI strategies geared towards the recognition and appreciation of different stakeholders;(2)To recognize the contributions made by migrant care workers and informal carers, and to define their role, including them in the LTC system, also as users of support services;(3)To invest in local projects that can be consolidated into innovative practices on a sustainable basis.

## 4. Discussion

The results of this study show that, potentially, a strong relationship can exist between SI and LTC in Italy, which possibly implies a mutual support between the two: the adoption of SI in LTC can lead to enhancement of the LTC system itself, while improvements in the LTC sector can promote SI in the wider context of welfare policies. When properly governed, the implementation of SI in LTC can represent therefore a structural factor to bring out an ongoing process of in-depth modernization of the welfare state, applying a new “ageing paradigm” based on the recognition of the positive contribution to society coming from older people and from the care provided to them [52,53].

In this regard, three macro issues were repeatedly mentioned by the respondents consulted as key components for a full implementation of SIs within the LTC sector: a focus on user needs; and both governance and management of care by a partnership of different stakeholders. Each of these factors found different forms of implementation in terms of SIs, key factors to promote SI, and recommendations. The promotion of SI is strongly related to an extended and innovative vision of the LTC system, supported by the holistic aim of improving the quality of life in old age. To be innovative, the Italian LTC system has to provide a new user definition that includes families, informal carers, migrant care workers and professionals. Each category of users has to be the beneficiary of care services and social support policies, to support individual well-being and utilize its contribution to the care provision path. Not accidentally, “case management” and “genuine attendance” were deemed innovations suitable for addressing user needs. Genuine attendance addresses the need to involve the appropriate services or care providers when considering individual care needs, while case management facilitates the provision of better care, by coordinating and integrating care workers, and by developing their specific skills. The “Up-Tech” and “Family nurse” good practices both move in this direction, albeit with a key difference: the former pushes the case management carried out by public institutions, while the latter represents a fully private solution, run by an NGO. 

A further, specifically innovative feature has also been identified in the policies addressing migrant care workers and informal carers, two target groups that have been largely neglected so far. In Italy, no national law acknowledges the existence of informal carers, apart from a very recent Bill passed by the Italian Parliament, which only indirectly recognizes their existence, by establishing a nationwide fund to support them in the 2018–2020 period. Regarding migrant care workers, who are often hired as undeclared helpers by families, a national programme—concluded in 2015—tried to promote regional and local experiences to provide employment services specifically addressed to them, and to support an upgrade of their expertise in care provision by means of specialized education [54]. Unfortunately, only in some regions—e.g., Piedmont—has this experience become a structural part of the LTC system. The innovative skills of migrant care workers were also highlighted by the literature, emphasizing their role in terms of a more integrated, better quality of care and life for both care recipients and carers [55,56].

In this context, it is not surprising that home care represents one of the main sectors in which SI finds its implementation in LTC. This is also confirmed by the attention given to multi-stakeholder partnerships, as a form of organizational and governance innovation. At a local level, multi-stakeholder networks mostly manage support policies provided at home. The optimization and enhancement of available resources is one of the main goals proposed by organizational and governance-focused innovations. In this regard, the crucial question is: “How can SI be structurally included in LTC policies?”. The answer seems to be the “agency for innovation” proposed by some experts, that works as a sort of “innovation engine”, implementing policies or making a contribution towards promoting a better collaboration among different stakeholders. 

The abovementioned importance attributed to the prevention issue supports the idea that the innovative thinking on the LTC system exceeds the concept of care, understood as health and social care. In this regard, the recommendation concerning the optimization of resources and redistribution of expenditure must take into consideration the issue of prevention and the innovations proposed in this regard. Moreover, the prevention idea proposed by the Italian case study has a dual aim: to prepare older people as to the possibility of becoming dependent; and to promote a cultural change at societal level concerning what it means to be older. 

Moreover, it should be underlined that new technologies (e.g., ICT) that are often defined as drivers of innovation by the literature [57] have only rarely been mentioned in the interviews and focus groups. When asked about a possible reason for this, the consulted experts suggested two possible reasons: on the one hand, the fact that evidence concerning the innovative contribution made by new technologies in personalizing home care is already known. On the other hand, they argued that technologies promote social innovation only if their application is included in a new system of care, which reorganizes care delivery according to a new model that is able to fully exploit the potential of new technologies in a more systematic manner. 

Across Europe, the relevance of home-based care was a core issue in several countries. In Spain, for example, the national reform on LTC adopted in 2007 promoted cash benefit schemes for informal carers and migrant care workers (MCWs), including coverage of their social security insurance [17]. The Spanish reform promoted a specific recognition of the family right to be considered part of care path, entrusting to the local level the responsibility of identifying the specific policies to improve the quality of care. By comparison, the results of our study show that in Italy the inclusion of MCWs in the LTC system is pursued by means of experiences aiming at the recognition of MCW’s skills, as an intermediate step needed to formalize their contribution to the overall care path. Similarly, the above analyzed case study of innovation in the field of case management underlines that social innovation in this area seems to move along this inclusive operational strategy. In other cases, the relevance of innovations oriented to improve the coordination of the variegated available offer (e.g., the multi-stakeholders’ networks or the innovation hub) confirms the overall Italian trend towards a more mixed care approach.

## 5. Conclusions

This study shows the potentially positive relationship that exists between SI and LTC in Italy, and what innovations are needed in this country to promote a virtuous circle between the two. The existence of common aims—i.e., the improvement of users’ quality of life and well-being, and the optimization of societal opportunities—characterizes this relationship, and has an impact on policies at macro, meso and micro level. 

At a macro level, SI is strongly identified as a regulatory tool for reforms to promote better governance and management of care services and initiatives. In the fragmented and decentralized Italian LTC system, innovative regulation should be promoted at regional or—as suggested by some experts—at national level, taking into account regional differences and responsibilities. However, any reform, to be successful, would have to adopt an open-minded view of what policy targets should be included, and on the definition of users. As for the targets, in Italy SI is mainly designed to improve care provided at home, understood both as formal and as informal care. Prevention—if included in reform attempts—expresses the SI power that stems from open-minded strategies to contrast the negative impact on older people’s quality of life of becoming dependent, or to change the prevailing culture approach with regard to how ageing is viewed. The SI aim of optimizing resources and opportunities is achieved by means of integration and coordination of policy and funding strategies. The innovative vision of who is a user considers as “direct beneficiaries” of policies not only the older people, but also the informal carers, migrant care workers and professionals, all as potential users of support or care services. In particular, the migrant care workers become a key part of SI, by means of their crucial contribution to the LTC system.

The building of new collaborations between different stakeholders—via multi-stakeholder networks—appears to be the main Italian strategy to promote social innovation. Already quite common at a local level, the multi-stakeholder network experiences promote a collaborative vision of welfare, based on the appraisal of complementary missions and actions to contrast the unmet needs. Moreover, the idea of a specialized agency to promote innovation underlines that, in Italy, SI is especially achieved at the meso level and that innovation is considered a driving force in developing the LTC system.

At a micro level, social innovations, such as the “case manager”, underline that the integration and coordination strategy promotes a better ability of services to address user needs, and an improvement in the quality of life among older people. 

Finally, it can be concluded that this study confirms existing European trends, by showing once again that home-based care is the main context in which SI should take place, but underlining at the same time that innovation in governance and organization plays a key role in its delivery, together with ICT. In order to benefit from the latter tool, however, organizations require an appropriate assessment so that this can become a driver of SI, without over-estimating its potential, given the multiple channels through which SI is achieved.

Moreover, the Italian case seems to suggest that also in other familial care regime countries social innovation should grow up along the axis turning around home-based care, involving both informal carers and MCWs. Furthermore, integration and coordination strategies represent crucial channels to promote social innovation, particularly where based on the involvement of different stakeholders, and through a better coordination between prevention and LTC policies.

Finally, some limitations should be considered with regard to this study. In particular, the lack of a direct involvement of representatives of families and of MCWs among those consulted for both interviews and focus groups. Their viewpoints have been partly represented by those provided by specific stakeholder associations or trade unions, which in Italy traditionally play an advocacy role including these actors. While this choice ensured the largest representation of stakeholders reflecting different perspectives, it inevitably shaped the discussion primarily around topics concerning the organization and governance of formal care services. Another limitation concerns the fact that the literature review might have also included other potential databases to obtain a more comprehensive evidence about the investigated topic (e.g., Web of Science or Pubmed), which might have offered better filters to obtain records than those used by Google Scholar. Similarly, basing the data collection on the results emerging from the MoPAct and Cequa projects only might have limited the range of experiences and practices considered for this study. Future efforts should therefore try to adopt a more systematic approach in terms of both literature review and the variety of cross-national projects to be considered (such as, for instance, those described in the repositories offered by Eurocarers [58] or by the ACTIVAGE project [59]. Moreover, future studies should focus the attention on how the SI key factors are implemented in Italy.

Despite these limitations, we believe that the findings presented here highlight unprecedented information with regard to potential strategies to promote social innovation in Italy, which might also prove helpful for other countries, especially if they share a similar familistic approach to LTC provision.

## Figures and Tables

**Figure 1 ijerph-17-02367-f001:**
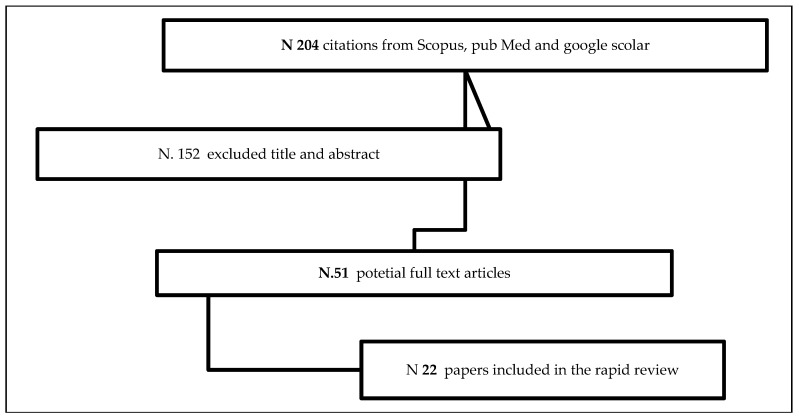
Flowchart of rapid review.

**Table 1 ijerph-17-02367-t001:** Typologies and number of stakeholders involved in the study via expert interviews and focus groups.

Typologies of Involved Stakeholders	Interviewed Experts (No.)	Participants in the Two Focus Groups (No.)	
		Ancona	Milan
Universities	4	0	1
National research institutes	2	1	2
Care providers (public and private profit services)	-	2	2
Policymakers	4	1	2
Nongovernmental organisations (NGOs) and foundations with an advocacy role for LTC recipients and their caregiving families	2	2	1
Social partners (trade unions/care professionals/Information and Communication Technologies-ICTs-industry managers)	-	3	2
**Total**	**12**	**20**	

Source: elaboration by authors.

**Table 3 ijerph-17-02367-t003:** Most urgent social innovations needed in Italy.

Categories	Social Innovations Needed in Italy	Description and Comments	Already Implemented
**Addressed to u sers’ needs**	Integrated mixed services to support the “family care and care worker”	System of services providing support to management of care workers. This system should provide support for: - selection of care workers, matching them with the appropriate families in need; - managing work contracts and other administrative aspects; - managing and supervising carers during care provision; - psycho-emotional counselling to family, carers and dependent older people; - training for carers.	Only at local level
	Case management	As a strategy to better integrate different care services. As a strategy to build a more personalized care path.	Only at local level
	Genuine admittance	As “true” taking on of patients/users; experiences exist in particular for specific issues such as for instance dementia.	Only at local level
**Innovative organizations and governance**	Multi stakeholders network	Collaboration between different stakeholders to plan, implement and governance of specific LTC initiatives. The contribution of each stakeholder is coming from his specific mission and point of view. The collaboration is usually supported by formal agreement acts that declare the common aims, the actions implemented and the single stakeholders roles. Moreover, one of stakeholder could be defined the “coordinator” to work as central governance body. In the most free form of network, the stakeholders agree on the general aims and collaborate without any formalized governance (e.g., coordinator).	Primarily at local level only.
	Innovation hub	This is an organism acting as an “attractor factor” of innovation. An idea or a starting project (or a set of ideas) around which interests can converge, regarding skills formation, structuring identity and developing activities. It could be different in different local contexts (local projects). To implement it, there must be an institutional setting able to support generative action, and/or a local coordinator/manager to identify resources and skills in the local community.	No
**Prevention**	Proactive prevention	Prevention activities to contrast the risk of becoming dependent. Support services to older people without dependency. Actions focused on services for social inclusion and wellbeing of older people.	No
	Educating about dependency	Prevention as an issue to be approached during the whole life: this means promoting the idea that older age is just one peculiar phase of life. A vision of community as an “educational” place: many activities can have these characteristics, in particular intergenerational activities.	No

**Table 4 ijerph-17-02367-t004:** SI Key factors in Italian LTC and specific elements.

Key Factors	Specific Elements
Coordination/integration	Initiative incorporates an integrated care model (e.g., health and social care; private and public care; formal and informal care).
Design	Design of the initiative is suited to meet target group’s needs (e.g., initiatives supported by need’s analysis study at local and national level, how had happen for the piedmont experience on recognition of informal).
Framework/structural conditions	Initiative draws on existing resources (e.g., human resources, existing built infrastructure, institutional resources and relationships, etc.). Overcome disinterest on the part of policymakers in Long term care as a political priority.
Funding	Raising private funds. Public sector co-financing. (e.g., specific integrated Funds at local and national level)
Leadership and governance of initiative	Institutional leadership of initiatives supported by not public institution (e.g., NGO).
Local/community focus	Initiative is adapted to meet local needs and contexts. Grassroots nature of the initiative/strong sense of community ownership.
LTC specificity	Initiative incorporates a community care model. Case management component is incorporated into the initiative’s care model. Initiative incorporates a user-centred care plan to meet individual needs.
Network	Well-established and active stakeholder network. Public-private partnership. Contributions of volunteers. Formal, legal foundation for institutional partnerships. Multi-actor/multi-sector cooperation.
Sustainability	Successful transition from pilot programme.
Workforce	Built-in element to ensure workforce sustainability (e.g., train-the trainer programmes).

Source: authors’ analysis based on [26].

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
