# Peer review of "Social Innovation in Long-Term Care: Lessons from the Italian Case"

_ijerph, 2020, doi:10.3390/ijerph17072367_

Round 1

Reviewer 1 Report

The focus of the research is of huge interest. However, the lengthy sentences and different terminology make it difficult to continue reading.  Using consistent and simple words or transitional words and phrase would enable readers to retain interests.

Specific suggestions are as follows:

  • Introduction section: please add the beneficial roles of SI for the LTC system in countries that focus on “familistic” care regime and decentralization governance model and cite scientific evidence to support.

Method section:

  • Please explain the use of “active ageing” as the search keyword for the literature review.
  • Please further elaborate the inclusion and exclusion criteria and indicate the number of papers/grey literature/policy documents/reports being selected for review/excluded and for final analysis based on these criteria.

Result section:

  • Please add the opinions given from the experts about the additional room for SI in the three selected SI case studies. Any additional ideas/opinions given from the experts beyond the key conditions described in Table 2?
  • Please elaborate the program detail for the Up-Tech cases in Table 2.
  • For the three selected cases, the authors used the 9 key conditions based on Heinze and Naegele’s framework to evaluate if the case is “good SI practices”. The authors then described the most urgently needed SI in Italy based on the seven innovations emerged from the experts’ opinions. A bit confused of what the concept/framework of SI adopted by the authors. Wonder if the seven innovation/three categories are under the 9 key conditions or these three categories are the additional key conditions emerged in your study?
  • Please elaborate the description of the specific elements of the ten key factors in Table 4. The specific elements of some key factors such as “leadership” is not clearly described. Also, it would be better to identify which key factors facilitate/hinder which key conditions/innovations.

Discussion section:

  • The authors described the “Up-Tech” and “Family nurse” programme with new terms such as “public governance of case management” and “social cooperative”. It would be better to use consistent terminology of the SI model/features.
  • In the last sentence about the technologies, please further elaborate the meaning of “only if their application is included in a new organization of care”?

Conclusion section:

  • The authors seems to summarize how the seven innovations impact the society at macro, meso, and micro level. Better to highlight this point in the introduction section as well. Wonder if these seven innovations are the SI framework suggested by the authors? How these seven innovations fit in the countries that focus on “familistic” care regime and decentralization governance model and the challenges of the implementation of these innovations.  Again, are the 9 key conditions and 7 innovations a "new" SI framework suggested by the authors?

Author Response

Dear Reviewer, many thanks for your in-depth reading and your useful suggestions to improve the paper. In the attached file you can find specific answers to each of your suggestions.

Reviewer 2 Report

Dear Authors,

I find the article an interesting read and appreciate the methods of obtaining qualitative data for coming up with the findings and the recommendations. However, I would assume that the one aspect that is missing in the paper is the family caregivers themselves since they represent the most important link in the older adult care paradigm.

I would appreciate if themes such as social innovation and further involvement into the all-pervasive familial caregiving structures would come forth based on actual family-caregiver and older-adult dyads such that actual care needs and likelihood of social support and governmental support apart from hospital/clinical care support can be established.

Reviewer 3 Report

This paper discussed a very important topic-social innovation in long-term care, which used data from the Italian case. In general, this paper is of good quality regarding its rich data source, wide range of research participants, clear presentation of the results and high relevance to social policy. It is well-written and certainly appropriate for the journal.

Readers may find this a worthwhile article. That said, however, this reviewer sees several places that need further attention from the authors.

Abstract

1. The authors may consider to add information about the research methods adopted in this study in the abstract.

2. Line 13, the word ‘ageing’ is British spelling, but some words in the main text, for example, in page 12, line 332, ‘realization’ and line 338, ‘utilize’ are American spelling. Please be consistent in spelling.  

3. line 23, ‘old age’, please double check and keep the spelling clean.

Introduction

There are also some typos:

1. page 2, line 58, there is a space between ‘Since’ and ‘the bulk of …’, please delete the space

2. line 60, the ‘]’ after the word ‘care’ needs to be deleted.

Material and methods

1. page 3, How long did the interviews and focus groups last, what are the time ranges?

2. The authors mentioned that the focus groups were performed in two cities, however, in Table 1, we cannot see the distribution of focus group participants in each research site. This information is important as the readers could see if there is potential sampling bias in participants selection.

3. Also in table 1, the ‘care providers’, are they referring to the migrant care works or family carers?

4. Page 3, line 120, a space in front of the word ‘What’ needs to be deleted.

5. In this section, the authors provided the information that related to data source and data collection, but did not illustrate clearly how they analysed the data. In qualitative research, there are a few methods for data analysis, such as ‘thematic analysis’, ‘Interpretative phenomenological analysis ’, ‘grounded theory’, ‘discourse analysis’, and so on. What method/methods was/were used in data analysis in this research, and what was the process of data analysis?

This information is helpful as it relates to the validity and creditability of this study. Please also provide information regarding research validity and creditability in the methods section.

Results

1. page 7, line 224, the number ‘3’ is missed

2. line 250, the word ‘a’ should be deleted.

3. Table 3, please delete the space between ‘selection of’ and ‘care workers’

4. ‘As “true” taking on of patients/users; experiences exist in particular for specific issues such as for instance dementia.’ Please delete ‘such as’ or ‘for instance’

Discussion

1. the second paragraph, line 329, please delete the space between ‘the’ and ‘respondents’

2. the next page, line 375, there is no subject for the sentence ‘have only rarely been mentioned in the interviews and focus groups.’

3. It is suggested to discuss about the generalisation of the results from this study to other countries or cultural contexts.

Reviewer 4 Report

It seems that it has been an ambitious work in which the authors have wanted to do many different things but without going into any of them in depth, such as: not having carried out a more meticulous review of the literature, and not having gone deeper into qualitative research.

In author contribution is explained that G.C. did the stadistical analysis, where is it? In material and methods authors indicated that is a qualitative analysis, and any stadistical were explained. The analysis process wasn't mentioned.

The literature review can be improved. The are potential databases to obtain better evidence about the topic as Web of Science, Pubmed as well, and others. Google is usefull as Google Scholar, but they don't have usefull filters to obtain good records.

It can be interesting to explore European project related to the topic. For example, look for Eurocarers ones, ACTIVAGE project and others.

Round 2

Reviewer 1 Report

The revised manuscript has been greatly improved although some sentences are still quite long. Minor spell check required and deleted the presence of (find?). Also, please spell out before using the abbreviated "MCWs".

Reviewer 4 Report

Page 6 appears with a blank hole and stripes.

The authors have integrated all the suggestions I have made.